# SUMMEDITS: Measuring LLM Ability at Factual Reasoning Through The Lens of Summarization

**Philippe Laban    Wojciech Kryściński    Divyansh Agarwal    Alexander R. Fabbri**
**Caiming Xiong    Shafiq Joty    Chien-Sheng Wu**
Salesforce AI
{plaban, divyansh.agarwal, afabbri, cxiong, sjoty, wu.jason}@salesforce.com

## Abstract

With the recent appearance of LLMs in practical settings, having methods that can effectively detect factual inconsistencies is crucial to reduce the propagation of misinformation and improve trust in model outputs. When testing on existing factual consistency benchmarks, we find that a few large language models (LLMs) perform competitively on classification benchmarks for factual inconsistency detection compared to traditional non-LLM methods. However, a closer analysis reveals issues with existing evaluation benchmarks, affecting evaluation precision. To address this, we propose a new protocol for inconsistency detection benchmark creation and implement it in a 10-domain benchmark called SUMMEDITS. This new benchmark is 20 times more cost-effective per sample than previous benchmarks and highly reproducible, as we estimate interannotator agreement at about 0.9. Most LLMs struggle on SUMMEDITS, with performance close to random chance. The best-performing model, GPT-4, is still 8% below estimated human performance, highlighting the gaps in LLMs' ability to reason about facts and detect inconsistencies when they occur.

## 1 Introduction

With recent progress in generation capabilities of LLMs, automatic summarization is making its appearance in practical information consumption situations such as summarizing work meetings (Arabzadeh et al., 2022), health records (Jain et al., 2022), or scientific documents (Cachola et al., 2020). To ensure the safe and effective implementation of these applications, it is essential to limit the reach of factually inconsistent summaries, a known issue with generated summaries (Kryściński et al., 2019; Maynez et al., 2020).

Prior work (Kryściński et al., 2020; Fabbri et al., 2021; Gao and Wan, 2022) has annotated corpora of model summaries with labels of factual con-

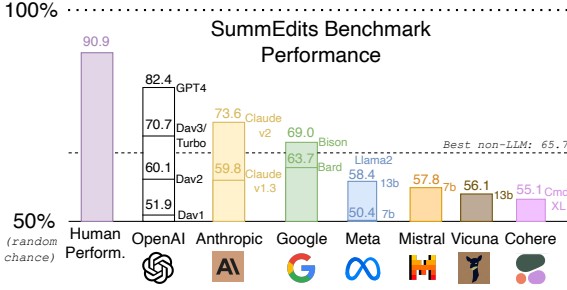

Figure 1: SUMMEDITS is a benchmark to evaluate the factual reasoning abilities of LLMs, measuring if models detect factual inconsistencies when they occur in summaries. Capable detection models can help build more reliable NLG systems.

sistency, finding that most abstractive summarization systems produce a non-negligible portion of inconsistent summaries. In turn, such corpora are used to instantiate tasks such as *inconsistency detection* (ID) (Laban et al., 2022a; Tang et al., 2022), in which models are given (`document,` `summary`) pairs, and must identify whether the summary is consistent with the document.

Recent investigations of using LLMs for evaluation have shown promising results across different NLP tasks (Liu et al., 2023; Fu et al., 2023), including factual consistency (Luo et al., 2023). In this work, we continue this line of research and explore applying LLMs as factuality evaluators in the context of text summarization. We first establish baseline performance for a suite of LLMs on two existing benchmarks – AggreFact (Tang et al., 2022) and DialSummEval (Gao and Wan, 2022) – with results confirming that some LLMs perform competitively with state-of-the-art specialized methods such as QAFactEval (Fabbri et al., 2022). However, a manual analysis of conflict cases in AggreFact reveals a significant number of mislabeled samples (7+%) of factual inconsistencies undetected by annotators during dataset creation that the GPT4 explanations reveal. This lack of quality of benchmarks lim-

its the precise evaluation of model performance at factual inconsistency detection.

To address this issue, we introduce a protocol designed to create challenging benchmarks while ensuring the reproducibility of the labels. The protocol involves manually verifying the consistency of a small set of seed summaries and subsequently generating numerous edited versions of these summaries. We discover that assessing the consistency of edited summaries is relatively straightforward and easy to scale for human annotators, thus guaranteeing low cost and high agreement among annotators, yet keeping the task challenging for models.

We create the SUMMEDITS benchmark by implementing the protocol in ten diverse textual domains, including the legal, dialogue, academic, financial, and sales domains. Figure 1 summarizes experimental results on the benchmark, which indicate that SUMMEDITS presents a challenge for all models, with only four LLMs outperforming the specialized model QAFactEval. Our estimate of human performance of 90%+ largely outperforms models, suggesting current LLMs are not yet proficient at complex factual reasoning, and cannot assess the factual validity of summaries with precision.

We believe SUMMEDITS can serve as a tool to evaluate LLMs' abilities to detect factual inconsistencies when they (inevitably) occur and encourage LLM developers to report their performance on the benchmark. For practitioners requiring specific domain expertise, the protocol can be adapted to generate low-cost, in-domain benchmarks that can check model capabilities prior to production use. We release the code and benchmark publicly[1].

## 2   Related Work

**Annotating Factuality of Summaries.**   With advances in language models and the increase in fluency and abstractiveness of summarizers, prior work showed that one of the key challenges in summarization was enforcing factual consistency (Kryściński et al., 2019), particularly with models trained on datasets with unfaithful references (Maynez et al., 2020). Several efforts – such as FactCC (Kryściński et al., 2020), SummEval (Fabbri et al., 2021), Polytope (Huang et al., 2020), FRANK (Pagnoni et al., 2021), and CLIFF (Cao and Wang, 2021) – annotated the generated summaries of tens of model, finding that most models produce a non-negligible portion of inconsistent summaries. Although most annotation effort has focused on the summarization of news, some prior work also looked at dialogue summarization (Gao and Wan, 2022), or the medical domain (Tang et al., 2023). In most work, scalable high-quality annotation is challenging, due to low inter-annotator agreement when relying on crowd-workers, with some work showing that 10+ annotators are required to achieve some level of consensus (Falke et al., 2019), and some work recommending solely relying on experts (Fabbri et al., 2021). At the heart of the issue, annotating the factual consistency of a summary is challenging: it requires careful reading of long documents and the detection and interpretation of nuanced facts. In this work, we propose a new protocol to annotate factual consistency resources and show that it lowers the cost and increases reproducibility by minimizing the amount of reasoning required for each annotation.

Some work has also annotated inconsistency of pairs on text on non-summarization tasks, such as paraphrasing (Zhang et al., 2019), document-grounded dialogue (Honovich et al., 2021; Dziri et al., 2022), and Wikipedia-editing (Schuster et al., 2021). Follow-up work has then aggregated and standardized annotations into benchmarks such as SummaC (Laban et al., 2022a), AggreFact (Tang et al., 2022) and TRUE (Honovich et al., 2022).

**Detecting Factual Errors.**   Some work has taken an automated approach to the detection of inconsistencies, with approaches falling into two main categories: question and entailment-based. In question-based approaches, questions are generated with the expectation that paired documents and summaries should provide consistent answers. QAFactEval (Fabbri et al., 2022) unified prior work (Wang et al., 2020; Scialom et al., 2021; Honovich et al., 2021) by systematically evaluating each element of the pipeline and proposing a best-performing combination. Entailment-based methods either rely on entailment on dependency parses – DAE (Goyal and Durrett, 2020) – or directly leverage natural-language entailment models, such as SummaC (Laban et al., 2022a). We include these three representative models in our experiments, finding that even with several orders of magnitudes fewer parameters than LLMs, they can reach similar performances on benchmarks.

---

[1] https://github.com/salesforce/factualNLG

# 3 Limits of Crowd-Based Benchmarks

We first analyze model performance on two popular benchmarks for factual consistency detection in summarization: AggreFact (Tang et al., 2022) and DialSummEval (Gao and Wan, 2022) and uncover limitations that guide the design principles of the SUMMEDITS benchmark.

## 3.1 Experimental Setup

We include in our experiments three specialized non-LLM approaches: DAE, SummaC, and QAFactEval and ten LLM models from recent LLM families. We include Cohere's Command-XL, Anthropic's Claude V1.3 (Bai et al., 2022), Google's Bard and PaLM2-Bison (Thoppilan et al., 2022), Vicuna-13b (Chiang et al., 2023), and OpenAI's DaVinci001 (Brown et al., 2020), DaVinci002 (Ouyang et al., 2022), DaVinci003, GPT3.5-turbo, and GPT-4. Appendix A provide each model's method of access and model card.

To minimize the computational cost of experiments, we select a single Zero-Shot prompt that is used for all LLM models. We make this choice instead of optimizing the prompt for each model for two reasons: (1) there's no guarantee that prompt quality will transfer across benchmarks, and using a single common prompt removes variance from prompt optimization that does not measure underlying model ability, and (2) more complex prompts would require adaptation to each domain (e.g. domain-specific few-shot examples), and restrict the evaluation of models with shorter maximum sequence lengths due to longer prompts.

## 3.2 AggreFact

AggreFact-SOTA (Tang et al., 2022) is a factual consistency benchmark focused on the news domain, modified from SummaC (Laban et al., 2022a) to focus on summaries generated by *SOTA* models (i.e., models based on pre-trained Transformers), as analysis showed older models' summaries were less relevant to the field of consistency detection.

Table 1 reports the balanced accuracy of specialized models and LLMs on AggreFact. At first glance, the specialized models still outperform LLMs, even though increasing LLM size leads to performance improvements and helps close the gap, with GPT-4 performing within 2.4% points of the specialized DAE. However, all models perform relatively poorly, with no model reaching a balanced accuracy of 80% on a binary classification task.

| Model Name | AggreFact | DialSummEval | |
| --- | --- | --- | --- |
| | %BAcc. | %BAcc. | Corr. |
| DAE | **76.0** | 56.2 | 0.44 |
| SummaC | 71.6 | 62.7 | 0.35 |
| QAFactEval | 73.9 | 64.4 | **0.59** |
| Cohere-cmd-XL | 63.1 | 56.6 | 0.36 |
| Claude V1.3 | 50.6 | 56.8 | 0.30 |
| Bard | 62.7 | 59.5 | 0.26 |
| PaLM2-Bison | 57.0 | 55.6 | 0.57 |
| Dav001 | 53.3 | 52.9 | 0.11 |
| Dav002 | 54.3 | 59.2 | 0.49 |
| Vicuna-13b | 60.3 | 58.6 | 0.36 |
| Dav003 | 64.8 | 60.9 | 0.51 |
| GPT3.5-turbo | 70.2 | 62.0 | 0.56 |
| GPT-4 | 73.6 | **68.4** | 0.58 |

Table 1: Performance of models on the AggreFact, DialSummEval consistency benchmarks reported in balanced accuracy (**%Bacc.**) and correlation (**corr.**).

To inspect performance on the AggreFact benchmark further, we hired an annotator to manually inspect the cases where GPT4 disagrees with the label of AggreFact. More precisely, we manually inspected the explanations provided by GPT4 for the 101 summaries it judged were inconsistent but labeled as consistent in the dataset. Appendix B provides more detail on the annotation protocol.

Of the 101 samples, 80 were labeled by the annotator as correct or partially correct explanations that identify and explain a factual inconsistency in the summary. In other words, this manual analysis of a subset of AggreFact reveals that **a minimum of 6% of the samples in AggreFact are mislabeled.** The low reliability of labels in crowd-sourced benchmarks like AggreFact is a known issue (Pagnoni et al., 2021) stemming from task complexity that requires the annotator to carefully read and understand an entire document and accompanying summary, leading to low repeatability and inter-annotator agreement.

This analysis reveals the potential for LLMs as part of dataset creation. In some cases, an LLM explanation that is verifiable – such as an explanation for an identified factual inconsistency – can accelerate and improve the quality of annotation. LLM explanations might not be valuable in all cases, such as when a model asserts a summary is consistent, manual verification is still required to assure quality. In Section 5, we introduce a protocol for benchmark creation that can involve an LLM.

Based on the low reliability of labels in AggreFact, we note that a key requirement for future

| | **Average Annotator Likert Score** | | | | | | | |
|---|---|---|---|---|---|---|---|---|
| **Model** | 1.5 | 2.0 | 2.5 | 3.0 | 3.5 | 4.0 | 4.5 | 5.0 |
| Dav001 | 68.1 | 78.4 | 84.6 | 90.2 | 83.6 | 84.9 | 86.0 | 88.9 |
| Cohere-cmd-XL | 46.2 | 51.0 | 70.3 | 83.6 | 88.6 | 89.2 | 91.7 | 96.3 |
| DAE | 30.8 | 56.9 | 63.7 | 83.6 | 86.8 | 94.3 | 90.3 | 94.2 |
| PaLM2-bison | 25.3 | 35.3 | 56.0 | 78.7 | 93.6 | 97.2 | 98.4 | 95.8 |
| Dav002 | 13.2 | 29.4 | 47.3 | 62.3 | 77.7 | 83.0 | 88.3 | 90.0 |
| Dav003 | 4.4 | 17.6 | 28.6 | 31.1 | 63.2 | 69.3 | 84.9 | 81.6 |
| GPT3.5-turbo | 8.8 | 15.7 | 29.7 | 45.9 | 73.6 | 76.4 | 88.5 | 90.0 |
| GPT4 | 2.2 | 5.9 | 6.6 | 24.6 | 45.9 | 54.2 | 80.9 | 87.9 |
| QAFactEval | 3.3 | 5.9 | 17.6 | 24.6 | 44.5 | 54.7 | 70.3 | 74.7 |
| Vicuna-13b | 8.8 | 15.7 | 17.6 | 37.7 | 50.9 | 54.2 | 65.5 | 66.8 |
| SummaC | 4.4 | 5.9 | 20.9 | 21.3 | 27.7 | 40.1 | 43.7 | 58.9 |
| Claude V1.3 | 1.1 | 9.8 | 11.0 | 13.1 | 33.6 | 37.3 | 47.1 | 45.8 |
| Bard | 9.9 | 7.8 | 5.5 | 9.8 | 18.2 | 21.2 | 36.5 | 42.6 |

Table 2: Percent of summaries classified as consistent in DialSummEval, bucketed by average Likert consistency score. Models interpret the Likert range differently.

benchmarks is to improve label reliability, which can be demonstrated with high annotator agreement when multiple annotators are involved.

### 3.3 DialSummEval

The DialSummEval (Gao and Wan, 2022) benchmark is a summarization evaluation benchmark created following the format of SummEval (Fabbri et al., 2021) for the domain of dialogue summarization. In DialSummEval, each (dialogue, summary) tuple is evaluated by three annotators, each assigning a Likert score (1-5) assessing the consistency of the summary. The authors of the benchmark report an agreement level of 0.67 Krippendorff's alpha on the labels, indicating a moderate amount of agreement among annotators.

We evaluate model performance in two ways: (1) correlation between model predictions and the average annotator score, and (2) we follow Laban et al. (2022a) to transform the annotation into a binary classification task, amenable to the balanced accuracy metric. Results summarized in Table 1.

Echoing results on AggreFact, increasing model size leads to minor performance gains, with most LLMs underperforming specialized methods. In absolute terms, all methods struggle to achieve strong performance, with accuracies all below 70%.

In Figure 2, we aggregate model predictions into 0.5-width buckets on the Likert scale. We find that most models achieve strong performance on non-borderline buckets ([1.0, 1.5], [1.5, 2.0], [4.0, 4.5], [4.5, 5.0]), assigning a vast majority of samples to the correct class (inconsistent for low buckets, consistent for high buckets). The borderline buckets ([2.0, 4.0]) however are less clear-cut: most mod-

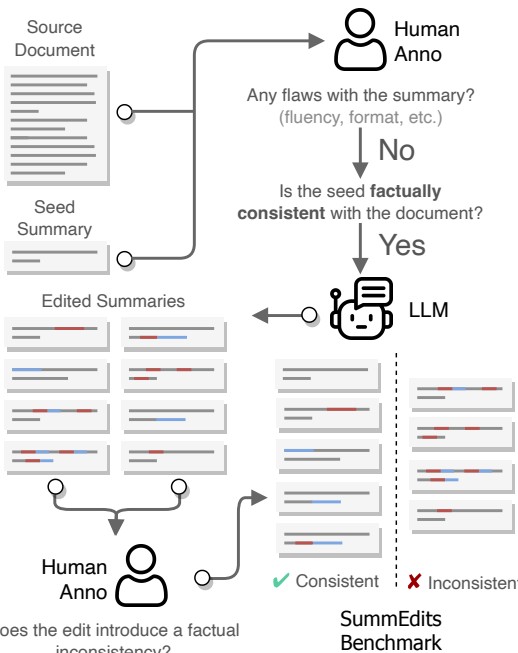

Figure 2: SUMMEDITS protocol diagram, a three-step protocol to create summarization ID benchmarks. See Table 3 for example samples produced by the protocol.

els assign large proportions of samples from each bucket into consistent and inconsistent classes.

We argue that **annotating the consistency of summaries using a Likert scale limits the quality and interpretability of the benchmark**, as it is not evident to interpret the differences between scores, limiting reproducibility, which is reflected in the moderate Kripendorff's alpha. Instead, we favor framing factual consistency benchmarks as a *detection task*. In the detection task, identifying any factual inconsistency between the document and summary leads to an overall assessment of the summary being *inconsistent*. If no inconsistency is detected, the summary is *consistent*. The detection framing also allows for models to provide natural language explanations when identifying a summary as inconsistent, which can be manually verified to confirm model reasoning ability.

In the next section, we propose a novel protocol to create factual consistency benchmarks, incorporating lessons learned from existing benchmarks.

## 4 SUMMEDITS Protocol

### 4.1 Design Principles

We set several design principles that help create higher-quality factual consistency benchmark:

P1. **Binary Classification Task:** In the bench-

| **Consistent** Edited Summary | **Inconsistent** Edited Summary |
|---|---|
| The characters discuss ponder the consequences of banishing Marcius, with Cominius warning that his alliance collaboration with the Volscians will bring great danger to Rome. | The characters discuss the consequences of banishing Marcius, with Cominius warning that his alliance with the Volscians Romans will bring great danger to Rome the Volscians. – `Entity Manipulation` |
| We introduced a novel new, simple, and efficient data augmentation method that boosts improves the performances of existing GANs when training data is limited and diverse. | We introduced a novel, simple, and efficient data augmentation method that boosts the performances of existing GANs when training data is limited abundant and diverse. – `Antonym Swap` |
| Employees of the European Commission are now forced instructed to delete remove TikTok from their work devices, and delete get rid of it from their personal devices too if they have work-related apps applications installed. | Employees of the European Commission are now forced not required to delete TikTok from their work devices, and delete but should still remove it from their personal devices too if they have work-related apps installed. – `Hallucinated Fact` |
| A conversation between a sales agent and a potential client possible customer. The sales agent provides information on different home insurance plans options and pricing, as well as available discounts for clients with good credit scores and other factors. | A conversation between a sales agent and a potential client. The sales agent provides information on different home insurance plans and, but not on pricing, as well as or available discounts for clients with good credit scores and other factors. – `Negation Insertion` |

Table 3: Example edited summaries – deletions, insertions – for from SUMMEDITS (domains top-to-bottom: Shakespeare Plays, SciTLDR, News, Sales Call). Inconsistent summaries are labeled with an `Edit Type`.

mark, a summary should either be labeled as inconsistent if any factual inconsistency is identified with the document or consistent otherwise, to improve label interpretability.

P2. **Focus on Factual Consistency:** Summaries in the benchmark should be flawless on aspects unrelated to consistency, such as fluency, coherence, and formatting, to avoid confounding effects on the quality of the benchmark.

P3. **Reproducibility:** Benchmark labels should not depend on annotator identity, and high annotator agreement should confirm the validity of the benchmark, as well as estimate human performance on the benchmark.

P4. **Benchmark Diversity:** Inconsistency errors in the benchmark should represent a wide range of errors in realistic textual domains, to increase understanding of model strengths and weaknesses, and better establish gaps in performance between models and human annotators at factual reasoning, if there are any.

## 4.2 Creation Procedure

We now describe the creation procedure for SUMMEDITS – illustrated in Figure 2 – which satisfies the design principles stated above.

The procedure consists of three steps: (1) seed summary verification, (2) generation of summary edits, and (3) annotation of edited summaries.

**Seed Summary Verification.** Benchmark creators select a small collection of documents in a domain of choice, and a *seed summary* for each document, which can be human-written or model generated. An annotator answers two questions about each (`document`, `seed summary`) tuple: (a) "Are there any flaws with the summary? (fluency, format, etc.)", (b) "Is the summary factually consistent with the document?". If the annotator identifies a flaw or an inconsistency, the tuple is filtered out (**P2**), otherwise, it proceeds to Step 2.

**Editing Summaries.** The second step consists in generating multiple *minor edits* of the summary, which might or might not affect the summary's consistency. This step can be carried out manually, or automatically with an LLM. Proposed edits should be atomic and localized, not entirely rewriting a novel summary. Table 3 gives examples of edits.

**Annotation of Edited Summaries.** The annotator who completed Step 1 reviews each edited summary, assigning one of three labels: (a) *consistent* if an edit does not lead to an inconsistency, (b) *inconsistent* if the edit modifies the seed summary in a way that introduces a factual inconsistency, (c) *borderline* for any other case such as the edit making the summary unclear, or requiring subjectivity.

Crucially, a single annotator should complete both Steps 1 and 3, as once they have invested the time in reading the (`document`, `summary seed`) tuple, judging the consistency of edits is a simpler task. We recommend including a large number of edits (e.g., 30 edits) to maximize edit diversity (**P4**) and encouraging annotators to assign the borderline label if they are unsure about any aspect of an edit to maximize reproducibility (**P3**).

A benchmark can be formed by retaining edited summaries that are labeled as consistent and inconsistent and filtering out borderline cases.

The procedure requires a small number of documents and seed summaries which are derived into many edited summaries. This flexibility facilitates the creation of factual consistency benchmarks in application domains that lack such resources.

| Domain | N | %Balance | IAA |
|--------|-----|--------|------|
| News | 819 | 39.2% | 0.91 |
| Podcast | 500 | 32.6% | 0.91 |
| Billsum | 853 | 42.3% | 0.90 |
| Samsum | 664 | 36.4% | 0.90 |
| Shakespeare | 814 | 46.4% | 0.96 |
| SciTLDR | 466 | 31.1% | 0.93 |
| QMSum | 431 | 42.5% | 0.92 |
| ECTSum | 668 | 38.0% | 0.96 |
| Sales Email | 613 | 29.2% | 0.87 |
| Sales Call | 520 | 33.3% | 0.93 |
| **Overall** | 6,348 | 37.10% | 0.92 |

Table 4: Statistics of the ten domains included in the SUMMEDITS benchmark, including the number of samples (**N**), the percentage of consistent summaries (**%Balance**), and the inter-annotator agreement (**IAA**).

## 5   SUMMEDITS Benchmark

### 5.1   Benchmark Creation

We implemented the SUMMEDITS protocol on ten realistic summarization domains to explore the reliability of the protocol. For five domains, seed summaries are automatically generated due to the lack or low quality of existing reference summaries. In such cases, we used GPT3.5-turbo and domain-specific prompts to generate seed summaries. We note that the quality of seed summaries is ultimately manually confirmed in step 1 of the protocol.

For all domains, we use GPT3.5-turbo[2] for Step 2. We experimented with integrating multiple LLMs in the edit generation process, but preliminary results indicated that many LLMs were not successful at generating minorly edited summaries and often attempted to write entirely novel summaries, which led us to solely use GPT3.5-turbo. More on this choice in Section 7.

We hired two professional annotators compensated at a rate of $20/hour to perform Steps 1 and 3. Three authors of the paper also participated in the annotation for quality control purposes. Appendix C has further detail on annotation protocol and an overview of the annotation interface. We next introduce the ten domains included in the SUMMEDITS benchmark.

**News**   To avoid selecting documents that are in the training corpora of evaluated models, we follow prior work (Goyal et al., 2022) and select `(document, summary)` tuples from re-

cent news articles. We obtained news articles from the Google News top events in February 2023, selecting at most one per news source to increase coverage diversity (Laban et al., 2023). Seed summaries are extracted from article metadata.

**Podcast (Clifton et al., 2020)**   We collected 40 podcast transcripts from Spotify's podcast summarization dataset's test set. We generated seed summaries due to low reference summary quality.

**BillSum (Kornilova and Eidelman, 2019)**   We collected 40 US bills and their summaries as seeds from the training portion of BillSum, a challenging dataset for summarization in the legal domain.

**SamSum (Gliwa et al., 2019)**   We collected 40 dialogues and summaries from the training portion of SamSum, a common dialogue summarization dataset for messenger-like conversations.

**Shakespeare (Karpathy, 2015)**   We collected 40 scenes from Shakespeare plays from the Tiny Shakespeare corpus, each roughly 700 words long. We generated seed summaries automatically.

**SciTLDR (Cachola et al., 2020)**   We collected 40 research paper abstracts and corresponding TLDRs from the training portion of SciTLDR, a dataset for scientific paper summarization.

**QMSum (Zhong et al., 2021)**   We collected 40 document and seed summaries from QMSum, a dataset for query-based meeting summarization.

**ECTSum (Mukherjee et al., 2022)**   We collected 40 documents from the ECTSum dataset, a summarization dataset for the financial earnings call transcripts. Due to low reference summary quality, we generated seed summaries automatically.

**Sales Call & Email**   We generated 40 fictional sales call transcripts, 40 sales emails, and corresponding seed summaries using ChatGPT. These domains evaluate the protocol's validity with entirely synthetic textual data in targeted domains that lack pre-existing summarization datasets.

### 5.2   SUMMEDITS Statistics

Table 4 provides statistics of the SUMMEDITS benchmark. Each domain yielded between 400-900 edited summaries, depending on the fraction of seed summaries that pass Step 1 (58% overall pass rate) and the percentage of edited summaries that are filtered out as borderline in Step 3 (around

---

[2]The prompts will be listed in our open-source release.

| Model | Podcast | BillSum | SAMSum | News | Sales C | Sales E | Shkspr | SciTLDR | QMSum | ECTSum | Avg. (↓) |
|---|---|---|---|---|---|---|---|---|---|---|---|
| DAE | 54.9 | 55.1 | 59.5 | 61.7 | 50.8 | 55.0 | 54.5 | 55.2 | 52.0 | 58.6 | 55.7 |
| SummaC | 58.5 | 55.7 | 54.7 | 62.1 | 59.0 | 57.7 | 59.3 | 59.7 | 56.6 | 64.4 | 58.8 |
| QAFactEval | 64.0 | 54.4 | 66.3 | 74.6 | 68.5 | 64.2 | 61.9 | 67.5 | 62.4 | 72.9 | 65.7 |
| Dav001 | 53.3 | 50.2 | 51.0 | 54.4 | 55.3 | 52.5 | 50.0 | 51.0 | 50.3 | 50.9 | 51.9 |
| Cohere-cmd-XL | 51.1 | 52.7 | 52.0 | 52.6 | 60.3 | 59.5 | 50.0 | 60.5 | 53.9 | 60.5 | 55.1 |
| Vicuna-13b | 52.8 | 52.6 | 50.8 | 63.0 | 58.1 | 51.8 | 55.5 | 59.7 | 54.0 | 62.5 | 56.1 |
| Claude v1.3 | 59.9 | 52.1 | 64.1 | 63.3 | 61.7 | 56.6 | 58.0 | 57.6 | 56.9 | 67.8 | 59.8 |
| Dav002 | 56.4 | 53.9 | 57.1 | 61.9 | 65.1 | 59.1 | 56.6 | 64.6 | 60.6 | 66.2 | 60.1 |
| Bard | 50.0 | 58.3 | 61.3 | 72.8 | 73.8 | 69.0 | 58.4 | 66.1 | 53.9 | 73.1 | 63.7 |
| PaLM2-bison | 66.0 | 62.0 | 69.0 | 68.4 | 74.5 | 68.1 | 61.6 | 78.1 | 70.2 | 72.3 | 69.0 |
| Dav003 | 65.7 | 59.9 | 67.5 | 71.2 | 78.8 | 69.4 | 69.6 | 74.4 | 72.2 | 77.9 | 70.7 |
| GPT3.5-turbo | 68.4 | 63.6 | 69.1 | 74.5 | 79.7 | 65.5 | 68.1 | 75.6 | 69.2 | 78.9 | 71.3 |
| GPT4 | 83.3 | 71.1 | 82.9 | 83.3 | 87.6 | 80.1 | 84.6 | 82.4 | 80.4 | 88.0 | 82.4 |
| GPT4 Oracle | 90.2 | 85.5 | 86.3 | 88.3 | 91.1 | 83.5 | 96.6 | 86.3 | 89.9 | 91.7 | 88.9 |
| Human Perf. | 90.8 | 87.5 | 89.4 | 90.0 | 91.8 | 87.4 | 96.9 | 89.3 | 90.7 | 95.4 | 90.9 |

Table 5: Balanced accuracy of models on the SUMMEDITS benchmark. Top three are non-LLM specialized models, middle section are LLMs, bottom section reports a GPT4 oracle performance and an estimate of human performance.

6%). In the five domains where seed summaries were generated by GPT3.5-turbo, 17.8% of the seed summaries were labeled as inconsistent, indicating that modern LLMs like GPT3.5-turbo struggle to remain consistent when summarizing documents.

For each domain, the seed summaries of at least ten seed summaries were annotated by multiple annotators, corresponding for each domain to at least 20% of the samples in the benchmark. In total, 1,419 of the 6,348 samples in SummEdits received multiple annotations, allowing us to measure agreement levels. When considering all three labels (consistent, inconsistent, borderline), Cohen's Kappa in each domain varies between 0.72-0.90, averaging 0.82. When removing samples annotated as borderline by any annotator, the average Cohen's Kappa rises to 0.92, **empirically validating the importance of filtering out borderline samples to create a reproducible benchmark.**

The edited summaries have and average of 3.6 words inserted, and 3.5 words deleted. These edit statistics do not vary widely based on the consistency label, as consistent edited summaries have an average of 3.6 words inserted, 3.7 words deleted, and inconsistent edited summaries have 3.6 words inserted, 3.4 words deleted. These statistics that models could not rely on structural signals to predict the consistency of a summary, and required factual reasoning to accomplish the task.

In the final benchmark, 37% of summaries are consistent, approaching our objective of a balanced benchmark to facilitate robust evaluation and minimize metric fluctuations (Luque et al., 2019).

The total annotation cost of SUMMEDITS is around USD 3,000, representing around 150 hours

of annotator work. The average cost of adding a domain to SUMMEDITS is around USD 300, within reach for NLP practitioners looking to evaluate the model ability in their domain of choice. Authors of the FRANK benchmark (Pagnoni et al., 2021) – samples of which are in AggreFact – estimate that each FRANK sample required 30 minutes of annotator time. At similar annotator pay, the annotation of a new domain of similar size to ones in SummEdits would cost an estimated USD 6,000: twenty times more. This cost analysis reveals the dual advantage of our protocol: by focusing the annotation task on atomic edits, the cost is drastically reduced and high reproducibility is maintained.

## 5.3 SUMMEDITS Results

Table 5 reports the performance of specialized models, LLMs with a zero-shot prompt, an oracle version of GPT4, and an estimate of human performance on the samples with multiple annotations.

Overall, model performance on the benchmark is low, with only GPT4 getting within 10% of human performance. Larger or more recent LLMs perform better on the benchmark, as is illustrated by the gradual improvements observed with each model generation in the OpenAI model family.

PaLM2-Bison, Dav003, ChatGPT, and GPT4 are the only four LLMs that outperform the best non-LLM approach QAFactEval, **providing evidence that most LLMs are not yet capable to reason out-of-the-box about the consistency of facts**.

All three specialized models achieve their highest performance in the news domain, unlike LLM models. The specialized models are likely calibrated to the news domain, which they are most fre-

| | **Inconsistent Edit Type** | | | |
| Model | EntMod | Anto | Hallu | Neg |
|---|---|---|---|---|
| DAE | 52.0 | 53.0 | 52.9 | 53.9 |
| SummaC | 56.8 | 56.8 | 55.3 | 57.3 |
| QAFactEval | 61.4 | 65.0 | 64.3 | 70.4 |
| Dav001 | 50.0 | 50.9 | 50.8 | 53.7 |
| Cohere-cmd-XL | 53.7 | 55.8 | 55.5 | 63.8 |
| Vicuna-13b | 55.2 | 57.1 | 56.2 | 61.0 |
| Claude v1.3 | 58.8 | 60.3 | 61.5 | 66.7 |
| Dav002 | 58.3 | 61.4 | 62.4 | 72.0 |
| Bard | 63.2 | 65.3 | 65.6 | 71.3 |
| PaLM2-Bison | 67.0 | 70.0 | 71.7 | 80.3 |
| Dav003 | 69.2 | 71.1 | 76.3 | 83.3 |
| GPT3.5-turbo | 70.7 | 70.6 | 74.2 | 79.7 |
| GPT4 | 82.2 | 81.3 | 87.0 | 92.7 |
| **Average** | 61.4 | 62.9 | 64.1 | 69.7 |

Table 6: Balanced accuracy of models on the SUMMED-ITS benchmark, broken down by type of factual error: Entity Modification (`EntMod`), Antonyms (`Anto`), Hallucination (`Hallu`) and Negation (`Neg`) insertion.

quently tested on (Goyal and Durrett, 2020; Laban et al., 2022a; Fabbri et al., 2022). This confirms the importance of creating multi-domain benchmarks to measure model ability in realistic scenarios.

Some domains such as Shakespeare's plays or the legal BillSum are more challenging to the majority of models, with the latter seeing no model score higher than 71.1%. Yet, factual reasoning in the legal domain is an important application area of NLP (Chalkidis et al., 2020; Shen et al., 2022).

We experiment with an oracle setting in which we append the seed summary to the end of the input document and input the concatenation to the model. The seed summary serves as an information scaffold, enabling the model to view modifications between the seed and edited summaries. GPT4 achieves a significant boost under the oracle setting, with the model performing within 2% of human performance. This confirms that high model performance on SUMMEDITS is attainable and that the challenge lies in aligning the facts of the edited summary with the document, without knowing that it has been edited.

### 5.4 Edit Type Analysis

We annotated each inconsistent sample in SUMMEDITS with tags of edit types.

The four edit types are: (1) *Entity Modification* in which an entity or phrase in the summary has been changed in a meaning-altering way, (2)

*Antonym Swap* when a word or phrase is replaced by a word of opposite meaning, (3) *hallucinated fact insertion*, when a novel fact is introduced in the summary which is not supported by the document, and (4) *negation insertion* when any negator word (e.g., not, neither) which modifies summary meaning is inserted. Figure 3 provides an example of each edit type in SUMMEDITS.

To annotate the entire benchmark, one author of the paper first manually annotated 200 samples of the dataset, which was used to evaluate several GPT4-based Zero-Shot and Few-Shot approaches. The best-performing prompt provides the definition of each edit type and a canonical example of each, and it achieved a performance of 0.85 F-1 and 0.92 recall, which was deemed sufficient for analysis purposes. GPT4 was used to annotate all inconsistent summaries in SUMMEDITS.

Overall, 78% of inconsistent summaries contain an entity modification, 48% an antonym swap, 22% hallucinated fact insertion, and 18% a negator insertion. The distribution of edit types is highly influenced by the LLM and prompt used to produce the edits in Step 2 of the protocol. Table 6 summarizes model performance by the edit type.

All models detect inconsistencies due to negator insertions the best, a sign that such errors are more discernable to models. Fact hallucinations are relatively harder to detect for non-LLM models but gradually become more evident to more performant LLMs. Finally, the entity modification and antonym error types generally see the lowest rate of detection by models across the board, perhaps due to such edits modifying an existing consistent fact in a more nuanced way.

### 5.5 Number of Edits Effect

In SUMMEDITS, it is common for the LLM to introduce multiple edits in each of its candidate summaries, as can be seen in the examples in Table 3, in which each edited summary contains multiple inserted and deleted words. In Appendix D, we analyze the effect of the number of edit types on model performance. In short, as the number of edit types in a summary increases, most models see sizable performance improvements, with average performance increasing from 59.2 to 74.1 when the number of edit types goes from 1 to 4.

This analysis confirms the perspective the task in SUMMEDITS corresponds to a *detection* task: as the number of introduced errors increases, model

performance increases as there is generally more evidence of inconsistencies for the models to detect. In turn, future work looking to create more challenging benchmarks using a similar protocol can focus on editing summaries with a single edit type.

# 6 Conclusion

In this work, we explore the capabilities of LLMs to act as factual reasoners through the lens of factual evaluation in text summarization. As part of this analysis, we uncover and discuss shortcomings of existing benchmarks. Using those insights we develop a new protocol for creating inconsistency detection benchmarks, which we implement in a 10-domain benchmark called SUMMEDITS. The SUMMEDITS benchmark is highly reproducible and more cost-effective per sample than previous benchmarks. Our experiments show that the benchmark is challenging for most current LLMs, with the best-performing model, GPT-4, still 8% below estimated human performance. We believe that SUMMEDITS can serve as a valuable tool for evaluating LLMs' abilities to reason about facts, detect factual errors and promote more reliable NLG systems. We encourage LLM developers to report their performance on the benchmark.

# 7 Limitations

**Why not fix existing benchmarks?** In Section 3, analysis reveals limitations with existing benchmarks that in theory can be fixed to yield improved versions of known benchmarks. The analysis we performed however only helps us invalidate a subset of samples in an opportunistic way, by looking at samples where benchmark labels and GPT4 disagree. However, this methodology cannot help us efficiently correct or confirm all samples, and improving existing benchmarks would require re-annotating a large portion of the benchmarks, and we do not have a guarantee that new annotations would improve on previous ones. By designing a new protocol for sample annotation that relies on clear, atomic edits, we simplify the annotation process, improving reproducibility.

**Effect of LLM in benchmark creation.** Step 2 of the protocol described in Section 4 relies on an LLM to generate many edits of the seed summary, which are subsequently manually annotated and included in the benchmark. The choice of LLM likely

has an effect on the benchmark which could favor a subset of LLMs most similar to the one used for benchmark creation. Initial attempts to use a pool of LLMs to produce edits were unsuccessful as we found that only ChatGPT and GPT4 were currently capable of following editing instructions that do not fully rewrite summaries. Future iterations on similar benchmarks should consider including diverse pools of LLMs in benchmark creation processes to avoid model-specific bias. Beside the edit summaries, we leveraged ChatGPT to generate the seed summaries in five of the ten domains in SUMMEDITS, due to the low-quality or non-existence of human-written summaries. All seed summaries are manually inspected by our annotators, and we did not find a gap in model performance dependent on the origin of the seed summaries.

**Beyond Binary Classification.** SUMMEDITS focuses on a binary classification formulation of factual reasoning (i.e., determining whether a summary is consistent/inconsistent). Binary classification has multiple advantages, including the ability to benchmark both generative and non-generative models, requiring limited adaptation of previous systems, and supporting well-established evaluation metrics such as balanced accuracy. However, the edit-based protocol of SUMMEDITS could be beneficial in instantiating more advanced factual inconsistency tasks. For example, SUMMEDITS could be modified into an "error localization" task which would require models to identify edit spans that render the summary inconsistent, or an "error correction" task, which would require a generative model to undo problematic edits, removing edit spans that lead to factual errors. These more advanced task formulations would require crafting reliable metrics, which was out of the scope of the current project.

**Evaluating Summarizers.** Previous annotation efforts in factual consistency of summarization were in part collected to evaluate which summarization models are least likely to generate factual inconsistencies (Falke et al., 2019). Since the summaries in SUMMEDITS are synthetic modifications of summaries, the benchmark cannot directly provide insights on summarizers and their ability to remain consistent, and the main purpose of SUMMEDITS is to measure LLM ability to reason about facts, and detect factual inconsistencies in text pairs. Future work could explore using meth-

ods such as Near-Negative Distinction (NND) (Laban et al., 2022b) to adapt SUMMEDITS into a set of tests to evaluate summarizer performance, and model ability to avoid generating inconsistent samples in the first place.

**Build Your Own Benchmark.** The initial release of SUMMEDITS consists of ten diverse domains we hope span common summarization domains. The current benchmark is however limited, as it only includes documents and summaries in English, and mostly limits document length to below 2,000 words. We have however shown that the protocol can be adapted to widely different textual domains – from US legal bills to Shakespeare plays – and produce domain-specific benchmarks at low cost. We hope that others will adopt and adapt the protocol to new domains, languages, and NLP tasks.

## Ethical Considerations

The models and datasets utilized in the project primarily reflect the culture of the English-speaking populace. Gender, age, race, and other socio-economic biases may exist in the dataset, and models trained on these datasets may propagate these biases. Text generation tasks such as summarization have previously been shown to contain these biases.

In Section 3 and Section 5, we recruited professional annotators to perform labeling with respect to summaries' factual consistency label or LLM reasoning explaining factual inconsistencies. We ensured to remunerate the participants fairly ($20/hour). Participants could communicate with us to voice concerns, could work at their own pace, and choose to stop working on the project at any time. Finally, we ensured to anonymize the annotations by not including personally identifiable information in any version of the dataset (annotator identity is instead marked as `annotator1`, `annotator2`, etc.).

In our work, we relied on several datasets as well as pre-trained language models. We explicitly verified that all datasets and models are publicly released for research purposes and that we have proper permission to reuse and modify the datasets.

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

## A  Model Access Detail

We experiment with a wide range of models. For each model, we specify its model card, and how it was accessed.

**Non-LLM models.** The three specialized models – SummaC[3], DAE[4], and QAFactEval[5] – were implemented through their online public repositories, and run locally on a multi-GPU machine (with 2 V-100 GPUs).

**Open-source Models.** We experimented with five open-source LLM models: LLama-13b (Touvron et al., 2023), Alpaca-13b (Taori et al., 2023), Dolly-V2-12b (`databricks/dolly-v2-12b`), Vicuna-13b (Chiang et al., 2023), and MosaicML's MPT-7b-chat (Team, 2023). All models were accessed through the public, online demonstration of LMSys.org[6]. Model responses were collected between April 15th, 2023, and May 15th, 2023.

---

[3] https://github.com/tingofurro/summac
[4] https://github.com/tagoyal/factuality-datasets
[5] https://github.com/salesforce/QAFactEval
[6] https://chat.lmsys.org/

**Google Models.** We experiment with two Google models, the Bard (Thoppilan et al., 2022) which we accessed through a web-based interface[7] which does not specify an exact model card, but model responses were collected between April 15th, 2023 and May 15th, 2023. Second, the PaLM-v2-bison model (Narang and Chowdhery, 2022) (model card `text-bison@001`), which was accessed through the Google Cloud VertexAI API.

**Anthropic Model.** We collected outputs of the Claude V1.3 model (model card: `claude-v1.3`), the latest and largest Anthropic model at the time of publication, using the official API hosted by Anthropic[8].

**Cohere Model.** We collected outputs of Cohere's `command-xlarge` model, the latest and largest Cohere model at the time of publication, using the official API hosted by Cohere[9].

**OpenAI Models.** We collected outputs for eight OpenAI models. Six models are from the GPT-3 family: Ada001 (`text-ada-001`), Bab001 (`text-babbage-001`), Cur001 (`text-curie-001`), Dav001 (`text-davinci-001`), Dav002 (`text-davinci-002`), and Dav003 (`text-davinci-003`). We also include GT3.5-turbo (`gpt-3.5-turbo`) and GPT-4 (`gpt-4`). All models were accessed through OpenAI's official API[10].

## B Explanation Annotation Guidelines

We hired a professional annotator to complete the annotation of model-generated explanations for AggreFact. The annotators were compensated at $20/hour. They received onboarding documentation that introduced them to the task, and provided the following definition for each type of explanation:

- **No Explanation**: If the model did not provide any explanation. (For example just saying: "The summary is inconsistent"),

- **Entirely Correct**: if the explanation correctly identifies and explains one or more factual inconsistencies in the summary,

- **Partially Correct**: if the explanation provided contains several elements and at least one of them correctly identifies and explains a factual inconsistency in the summary,

- **Unrelated**: if the explanation given does not directly relate to a factual inconsistency between the summary and the document,

- **Incorrect**: if the explanation given does not correctly identify a factual inconsistency in the summary, for example, making a logical error.

An example for each type of explanation was provided during onboarding. Annotation was performed in batches, and the first two batches of annotation by the annotator were reviewed by the authors of the paper. Incorrect annotations were discussed, allowing the annotator to better understand edge cases of the task, and modify their annotation in the first batches. Each annotator could communicate with one of the authors to discuss edge cases and maintain a common understanding of the task. Annotators could not communicate with each other.

## C SUMMEDITS Annotation Guidelines

We hired two professional annotators to complete the annotation of Steps 1 and 3 of the SUMMEDITS protocol (see Section 4). The annotators were compensated at $20/hour. They received onboarding documentation that introduced them to the task and used the interface shown in Figure 3.

Annotators were first assigned 10 warm-up seed summaries, each with roughly 30 edited summaries, which had been pre-annotated by the authors of the paper. The authors reviewed the completed warm-up exercises, and a strong agreement level on the warm-up task with both annotators was observed. Annotators could communicate with one of the authors of the paper to discuss any edge case or domain-specific question. For example, the annotation for the QMSumm domain required additional instructions due to query-focused formulation of the task, and instructions were communicated on how to deal with the "query" element when evaluating summaries. Namely, during Step 1 of the protocol, participants were asked to additionally judge whether the summary accurately responded

---

[7]https://bard.google.com/
[8]https://github.com/anthropics/anthropic-sdk-python
[9]https://docs.cohere.com/docs/the-cohere-platform
[10]https://github.com/openai/opeai-python

**Document:**

Simulation is a useful tool in situations where training data for machine learning models is costly to annotate or even hard to acquire. In this work, we propose a reinforcement learning-based method for automatically adjusting the parameters of any (non-differentiable) simulator, thereby controlling the distribution of synthesized data in order to maximize the accuracy of a model trained on that data. In contrast to prior art that hand-crafts these simulation parameters or adjusts only parts of the available parameters, our approach fully controls the simulator with the actual underlying goal of maximizing accuracy, rather than mimicking the real data distribution or randomly generating a large volume of data. We find that our approach (i) quickly converges to the optimal simulation parameters in controlled experiments and (ii) can indeed discover good sets of parameters for an image rendering simulator in actual computer vision applications.

**Original Summary:**

We propose an algorithm that automatically adjusts parameters of a simulation engine to generate training data for a neural network such that validation accuracy is maximized.

## Task 1:

Is any of the information in the summary **not** present in the document?
○ Yes  ◉ No

Are there any other issues with the summary? (incomplete sentence, formatting, etc.)
○ Yes  ◉ No

[Submit]

## Task 2:

Modified Summaries:

We propose an algorithm that automatically adjusts parameters of a simulation engine to generate training data for a neural network such that validation accuracy is maximized only slightly improved .
○ Inconsistent  ○ Consistent  ○ Borderline

We propose an algorithm that automatically adjusts parameters of a simulation engine to generate training testing data for a neural network such that validation accuracy is maximized.
○ Inconsistent  ○ Consistent  ○ Borderline

We propose an algorithm that automatically adjusts changes parameters of a simulation engine to generate training data for a neural network in such a way that validation accuracy is maximized.
○ Inconsistent  ○ Consistent  ○ Borderline

Figure 3: Two-column annotation interface used to annotate samples in the SummEdits benchmark. Participants could read the document on the left-hand column. Once they completed Task 1 in the right-hand column, the second annotation task became visible.

| Model | #Distinct Edit Types | | | |
|---|---|---|---|---|
| | 1 | 2 | 3 | 4 |
| DAE | 50.2 | 53.5 | 55.4 | 64.9 |
| SummaC | 58.2 | 56.3 | 57.6 | 67.3 |
| QAFactEval | 59.4 | 63.7 | 72.3 | 76.5 |
| Dav001 | 50.0 | 50.5 | 53.9 | 63.1 |
| Vicuna-13b | 52.8 | 57.0 | 60.2 | 58.5 |
| Cohere-cmd-XL | 50.0 | 55.9 | 63.7 | 70.0 |
| Claude v1.3 | 57.5 | 60.6 | 65.4 | 64.3 |
| Dav002 | 56.3 | 61.2 | 69.4 | 81.7 |
| Bard | 61.0 | 64.9 | 72.4 | 73.4 |
| PaLM2-Bison | 66.1 | 69.5 | 79.6 | 69.4 |
| ChatGPT | 68.5 | 71.4 | 82.0 | 86.6 |
| Dav003 | 65.3 | 72.0 | 85.8 | 88.8 |
| GPT4 | 81.0 | 83.0 | 92.0 | 94.3 |
| **Average** | 59.2 | 62.5 | 69.2 | 74.1 |

Table 7: Relationship between the number of edits types in the summary and balanced accuracy of models on SUMMEDITS. Models generally perform better as the number of introduced edits in a summary increases.

to the query, and otherwise mark summaries as inadequate.

## D Number of Edits Effect

Using the labels of edit types generated in Section 5.4, each edited summary labeled as inconsistent receives between one and four edit types. We group the summaries based on the number of distinct edit types that they contain, and report results on this axis in Table 7.

In general, we find that as the number of edit types present in a summary increases, the majority of detection models (both LLM and specialized) see sizable performance improvements, with an average performance of 59.2 for summaries containing a single error type, compared to 74.1 for summaries containing all four error types.