# OpenReview forum: "SummEdits: Measuring LLM Ability at Factual Reasoning Through The Lens of Summarization"
_EMNLP/2023/Conference — EMNLP 2023 Main_

### Official Review · Reviewer_jN9Y · 2023-08-04

**Soundness:** 4

**Excitement:**

4: Strong: This paper deepens the understanding of some phenomenon or lowers the barriers to an existing research direction.

**Missing References:**

* [Q2: : Evaluating Factual Consistency in Knowledge-Grounded Dialogues via Question Generation and Question Answering](https://aclanthology.org/2021.emnlp-main.619) (Honovich et al., EMNLP 2021)

* [TRUE: Re-evaluating Factual Consistency Evaluation](https://aclanthology.org/2022.naacl-main.287) (Honovich et al., NAACL 2022)

**Paper Topic And Main Contributions:**

This paper introduces a benchmark for measuring LM ability to detect factual (in)consistency in summaries. The authors construct this benchmark through a dual LLM, human pipeline wherein they select a set of summaries that annotators judge to be high quality (fluent, coherent, etc), ask an LLM to create several edited versions of the summary, and ask the same annotators to mark these edited summaries as factually (in)consistent. They perform this process for summaries in 10 domains. They assess a variety of LLMs and smaller models on the benchmark and find that smaller models trained specifically for factual consistency evaluation tend to outperform all but the largest models, while no model performs better than estimated human performance.

**Questions For The Authors:**

1. How different from the original summary are the edited summaries on average? Was there a threshold above which you would reject an edited article for being to different?
2. Can you describe the process of estimating human accuracy in more detail? Specifically, the proportion of data points that were used for each domain
3. What was the prompt given to the oracle version of GPT-4? Was it the same prompt as the other models but with the original summary added, or was there additional instruction?
4. Why did you try to use data that would not have been seen in training for news data but not for any other domains?
5. Do you assess other qualities of the edited summaries? Is it possible that there are differences in coherence/fluency/relevance between consistent and inconsistent versions and this is what models pick up on?
6. Relatedly, do you measure the average number of edits made to inconsistent models vs consistent summaries? Could it be that edits are identifiable and models have learned that consistent versions have more/less edits?

**Reasons To Accept:**

Detecting factual inconsistency is becoming a more and more important task, and a benchmark to measure model ability at this would be a valuable contribution. While this is proposed specifically with LLMs in mind, smaller, more specialized approaches could also benefit from this dataset.

The initial quality filter that removes summaries that annotators judge to have fluency/coherence issues is a good step, and the annotation procedure appears sound for the most part.

There are lots of model comparisons from models of different families and sizes indicating that this dataset is sufficiently challenging to be informative.

While I have some questions and concerns about various aspects of the paper, if the authors are able to adequately address these points, I would be inclined to raise my scores.

**Reasons To Reject:**

This work should include a comparison to the TRUE dataset (Honovich et al., 2022), which aggregates factuality assessment datasets across summarization and other text generation tasks.

I do find some issues in the annotation of the dataset. The entire dataset was annotated by only two people who were allowed to communicate both with each other and the authors of the paper during annotation. The reported inter-annotator agreement is quite high, but because of this, these scores are likely to be inflated due to communication between annotators and between annotators and authors throughout the process. It would have been a better practice to employ more than two annotators and not allow them to communicate with each other in order to get more accurate inter-annotator agreement scores. The authors argue that the high inter-annotator agreement scores indicate good potential for reproducibility and extending this task, but they may actually only indicate this if future annotators are also allowed to talk to each other and the authors. While the annotation questionnaire details appear clear to me, there is also no information on what criteria the authors applied to select qualified annotators or where annotators were hired from.

**Reproducibility:**

3: Could reproduce the results with some difficulty. The settings of parameters are underspecified or subjectively determined; the training/evaluation data are not widely available.

**Reviewer Confidence:**

3: Pretty sure, but there's a chance I missed something. Although I have a good feel for this area in general, I did not carefully check the paper's details, e.g., the math, experimental design, or novelty.

---

> ### Author Rebuttal · Authors · 2023-08-28
>
> We thank the reviewer for their in-depth review of our work.
> We agree with the reviewer that the inclusion of TRUE as a comparison to SummEdits would be beneficial to the readers, and will add a Section comparing SummEdits data statistics to previous benchmarks (SummEval, TRUE), offering a comparison in terms of size, domains and task covered.
>
> Regarding the protocol and annotation procedure, we thank the reviewer for their careful readthrough, and have the following comments:
> - We used ambiguous wording when describing the communication protocol between annotators and authors of the paper: “The annotators were added to a Slack channel with one of the authors”. More precisely, each annotator was added to an *individual conversation* with the first author of the paper, and not to a common Slack channel (the term Slack channel here was used incorrectly). The two annotators we hired did not know each other and did not communicate, and were hired on the professional recruiting platform UserInterviews. We agree that the two annotators communicating during annotation would undermine the reliability of the results, and confirm that this was not the case.
> - Besides the two annotators and the paper’s first author, two additional authors participated in the annotation which was used in the agreement level analysis. These two authors did not communicate with the professional annotators.
> - We believe these details were not clear in our current draft and will be corrected and expanded to include the details listed above.
>
> Question 1: We computed that on average, edited summaries have: 3.64 words inserted, and 3.48 words deleted. When breaking down by label: (1) for consistent edited summaries (3.69 words inserted, 3.69 words deleted), (2) for inconsistent edited summaries (3.60 words inserted, 3.35 words deleted). These statistics show that the summaries are minorly edited and that the amount of editing is roughly equivalent in consistent and inconsistent summaries. We will add these statistics to the dataset description section. We did not have a strict maximum number of words to be added or deleted but filtered out samples generated by the LLM that had a char-level Levenshtein string alignment of less than 0.7 to the seed summary, as we found that these were typically entirely rewritten summaries, rather than edited summaries based on the seed summary. We will add this detail in the procedure description.
>
> Question 2: To estimate the agreement level within each domain, samples of ten to thirty seed summaries, and all their accompanying edited summaries were doubly annotated, corresponding to at least 20% of the samples in each domain. In total, 1,419 of the 6,348 samples in SummEdits (~22%) received multiple annotations and were used to compute the agreement levels presented in Table 4.
>
> Question 3: Regarding the prompt of the oracle GPT4 model, we used the same prompt, and concatenated the seed summary to the document. This ensured that the model had access to the information in the seed summary, and could more easily view the edits and potential inconsistencies with the edited summary. We agree with the reviewer that this detail is lacking from the draft, and will add it. We will also note that the prompt could be modified to explicitly differentiate between the document and the seed summary, which we did not attempt, but could yield additional gains in performance.
>
> Question 4: In the news domain, the timestamp of documents allows for a simple way to control whether LLMs are likely to have seen the data during training, and since we could control this aspect, we ensured to use recent articles. For other domains (such as Shakespeare) it is likely that the models are trained on portions of the documents in SummEdits, and that the models know the plots of Shakespeare's plays, which likely affects their performance positively, but since we do not have access to exact training corpora of many LLMs, it is challenging to assess exactly. We will add this justification to the domain selection section.
>
> Question 5: In Step 1 of the process, annotators verify that the summaries do not have any flaws related to fluency, formatting, coherence, etc. However, we do not perform such a check on each edited summary. We instead rely on the fact that edits to the seed summary are minor and modify on average 3-4 words in the summary. Since the seed summary does not exhibit issues in fluency, formatting, or coherence, the edited summaries typically do not either. From the annotations performed by the authors of the paper, none of the edited summaries (either consistent or inconsistent) exhibited a lack of fluency, coherence, or issues in formatting, and LLM is instructed to target its edits to affect the consistency of the summary. We will add this explanation to the benchmark creation procedure.
>
> Question 6: See reply in Question 1. On average, consistent and inconsistent edited summaries see almost identical numbers of words inserted or deleted from the original seed summary. Therefore, it is unlikely that the model could leverage a difference in editing amount to improve performance. Furthermore, none of the models were trained on SummEdits, and therefore could not have learned patterns within the benchmark. However this is indeed a confounding factor that could affect the validity of the benchmark, and we will add the statistics to confirm that such bias is absent from the benchmark.

---

### Official Review · Reviewer_hL41 · 2023-08-05

**Soundness:** 4

**Excitement:**

3: Ambivalent: It has merits (e.g., it reports state-of-the-art results, the idea is nice), but there are key weaknesses (e.g., it describes incremental work), and it can significantly benefit from another round of revision. However, I won't object to accepting it if my co-reviewers champion it.

**Paper Topic And Main Contributions:**

This paper presents a new benchmark to test the ability of LLMs in detecting factual inconsistencies, which is more challenging and is 20 times more cost-effective per sample than previous benchmarks.

**Reasons To Accept:**

- Clear explanation of the benchmark creation process. The paper is easy to follow.
- Solid experiments which comprehensively compare the performances of most of the current popular LLMs.
- High reproducibility and diversity, which is crucial for creating a fair and generalized benchmark.

**Reasons To Reject:**

- The benchmark mainly uses binary classification task to measure the LLMs. I am curious that whether there are other more challenging tasks can also be integrated into the benchmark to further improve the diversity.

**Reproducibility:**

4: Could mostly reproduce the results, but there may be some variation because of sample variance or minor variations in their interpretation of the protocol or method.

**Reviewer Confidence:**

2: Willing to defend my evaluation, but it is fairly likely that I missed some details, didn't understand some central points, or can't be sure about the novelty of the work.

---

> ### Author Rebuttal · Authors · 2023-08-28
>
> We thank the reviewer for taking the time to review our work.
> Regarding the use of mainly a binary classification task to measure LLM performance, we agree that this is indeed an important choice in our work, which has implications.
> There are indeed other related tasks in summarization factual consistency which might be more complex and offer insights into LLM abilities, these include (a) error localization (specifying which portion of a summary contains a factual inconsistency), (b) error explanation/justification (in which a model would generate a free-text explanation for a factual inconsistency, verifying not only that the model’s label is correct, but that the model is able to explain what the error is), (c) error correction (in which a model might rewrite inconsistent summaries into factually consistent summary while minimally modifying the content).
>
> The three tasks described above are all more advanced than the binary classification task we implemented in our work, but the choice to focus on classification offers three main advantages:
> 1. Using a classification task allows us to compare generative models (LLMs) and non-generative models (SummaC, DAE and QAFactEval), which offers a wider range of comparison, and helps us understand how LLMs compare to specialized models (and in particular find that most LLMs lag these specialized models). The specialized models are not configured to perform the more advanced task formulations.
> 2. Classification comes with well-established metrics (Balanced Accuracy in our case), which make the future evaluation of models automated and reliable. More advanced tasks are more complex to evaluate, and could require manual annotation in the evaluation (for instance in the error explanation/justification task), which would limit the usability of the benchmark. Because we want to build a reusable benchmark, proposing an automatic and high-quality evaluation to accompany the benchmark is important.
> 3. Because binary classification has been the most common task formulation in inconsistency detection in summarization, it will allow more researchers in the community to test their methods on our benchmark without requiring adaptation, making the benchmark more accessible.
>
> Despite the fact that we focus on a relatively simple task formulation, we find that many models (including modern LLMs) achieve close to random performance and that there remains a large gap between human performance and top-performing LLMs, providing evidence that the SummEdits benchmark is challenging.
>
> We believe however that the edit-based protocol of SummEdits could be beneficial to instantiate more advanced factual inconsistency tasks. For example, SummEdits could be reused in an “error localization” task in which the model would have to identify edit spans that led to the inconsistency (since these are already labeled), or for an “error correction” task, in which one could observe whether the models remove the edit spans that led to an error. This would likely require the crafting of reliable metrics, which was out of the scope of the current project. We will add a paragraph to our Discussion to relay this future direction, and discuss more advanced task formulations that could be implemented based on the SummEdits benchmark we release.

---

### Official Review · Reviewer_eGPn · 2023-08-05

**Soundness:** 3

**Excitement:**

3: Ambivalent: It has merits (e.g., it reports state-of-the-art results, the idea is nice), but there are key weaknesses (e.g., it describes incremental work), and it can significantly benefit from another round of revision. However, I won't object to accepting it if my co-reviewers champion it.

**Missing References:**

https://arxiv.org/pdf/2302.04434

**Paper Topic And Main Contributions:**

This paper proposes a new protocol for inconsistency detection benchmark creation with a 10-domain benchmark called SUMMEDITS. The protocol involves manually verifying the consistency of a set of seed summaries and subsequently generating numerous edited versions of these summaries. This new benchmark also has high annotator agreement and is reproducible.

**Questions For The Authors:**

Provide some further discussion on how this work is more resource effective than fixing existing benchmarks that have been validated for annotator labels.

**Reasons To Accept:**

SUMMEDITS presents a challenge for all models evaluated, with only four LLMs outperforming the specialized model, and human performance outperforming all models. This analysis reveals the potential for LLMs as part of dataset creation and to identify factual errors.

**Reasons To Reject:**

The LLM used to generate seed summaries could influence the objectivity of the benchmark as an evaluation tool.

**Reproducibility:**

4: Could mostly reproduce the results, but there may be some variation because of sample variance or minor variations in their interpretation of the protocol or method.

**Reviewer Confidence:**

3: Pretty sure, but there's a chance I missed something. Although I have a good feel for this area in general, I did not carefully check the paper's details, e.g., the math, experimental design, or novelty.

---

> ### Author Rebuttal · Authors · 2023-08-28
>
> We thank the reviewer for taking the time to review our work.
> Regarding the use of a single LLM to generate seed summaries. We did not use LLM-generated seed summaries for all of the 10 domains in the SummEdits benchmark.
> - We prioritized the use of human-written reference summaries, which we used for five of the domains in the benchmark. For the other 5 domains, there were no high-quality reference summaries available, and we used GPT3.5-turbo to generate the seed summaries. We note that all seed summaries were manually inspected by annotators in Step 1 of the protocol, and low-quality summaries were filtered out.
>
> - Our experiments did not reveal that this choice of auto-generated seed summaries in some domains led to qualitative differences in the benchmark: for instance, one could have expected that LLMs would perform better on domains where seed summaries were auto-generated, but this in not observed in Table 5. For this reason, we decided to maximize domain coverage of the benchmark and include domains of interest to the summarization community (Sales Call, Creative Domain) by leveraging an LLM to generate seed summaries.
>
> - We will add a paragraph on this point in our Discussion, and direct participants to use the five domains with human-written references if they want to remove auto-generated seed summaries from their analysis.
>
> - Another possible observation could have been that the model used to generate seed summaries (GPT3.5-Turbo in our case) would have an unfair advantage in the benchmark. Although we do not have a way to measure whether a model is advantaged under the hood, we do observe that some models are able to outperform GPT3.5-Turbo performance on the benchmark, simply due to better factual reasoning abilities (for example GPT4). Since the original submission, we have conducted experiments with LLMs that have been published since then, and have found that other models (such as Claude 2.0) are able to outperform GPT3.5-Turbo and achieve performance only a few points below GPT4. This new finding confirms that models can achieve strong performance on the benchmark based on their factual reasoning abilities, even if they were not used in the benchmark creation process. We will add this new finding to the paper, and provide these discussion points.
>
> Regarding the resource effectiveness compared to fixing existing benchmarks. This point is discussed in the “Why not fix existing benchmarks?” paragraph of the Discussion. In short, the LLM-enabled analysis we performed in Section 3 to correct AggreFact samples can only be applied to a subset of the samples (samples where an LLM disagrees with the reference label), and cannot be used to fix existing benchmarks holistically. Therefore, fixing existing benchmarks would require re-annotation (which could be ~20x costlier per sample than our protocol). Another limitation with fixing existing benchmarks is that they often rely on summaries from old models (such as Pointer Generator) which make easier-to-detect mistakes. In essence, fixing existing benchmarks would be costly, and would not necessarily yield a challenging benchmark for the community, and we opted to create a new benchmark that is effective in terms of cost, and challenging to current detection models.
>
> We will include the missing reference provided by the reviewer, as it is related and relevant to the benchmark protocol we proposed.

---

### Meta-Review · Area_Chair_2E5G · 2023-09-18

**Recommendation:** 4

**Metareview:**

This paper creates a new benchmark dataset for assessing factual accuracy in summarization. They show that the task is challenging for LLMs.

Both reviewers hL41 and jN9Y agree on the soundness of the paper and the authors have addresses or promised to address the comments. In the rebuttal the authors seem to have addressed the concerns of eGPn.

This resource seems valuable to the community and could be quite impactful in judging new LLMs.

---

### Decision · Program_Chairs · 2023-10-07

**Decision:**

Accept-Main

**Comment:**

This paper creates a new benchmark dataset for assessing factual accuracy in summarization. They show that the task is challenging for LLMs.

Both reviewers hL41 and jN9Y agree on the soundness of the paper and the authors have addresses or promised to address the comments. In the rebuttal the authors seem to have addressed the concerns of eGPn.

This resource seems valuable to the community and could be quite impactful in judging new LLMs.